# Sex-Dependent Transcriptional Changes in Response to Stress in Patients with Myalgic Encephalomyelitis/Chronic Fatigue Syndrome: A Pilot Project

**DOI:** 10.3390/ijms241210255

**Published:** 2023-06-17

**Authors:** Jackson Gamer, Derek J. Van Booven, Oskar Zarnowski, Sebastian Arango, Mark Elias, Asha Kurian, Andrew Joseph, Melanie Perez, Fanny Collado, Nancy Klimas, Elisa Oltra, Lubov Nathanson

**Affiliations:** 1Institute for Neuro-Immune Medicine, Dr. Kiran C. Patel College of Osteopathic Medicine, Nova Southeastern University, Fort Lauderdale, FL 33328, USA; jg3380@mynsu.nova.edu (J.G.); nklimas@nova.edu (N.K.); 2Dr. Kiran C. Patel College of Osteopathic Medicine, Nova Southeastern University, Fort Lauderdale, FL 33328, USA; oz27@mynsu.nova.edu (O.Z.); sa2259@mynsu.nova.edu (S.A.); me1111@mynsu.nova.edu (M.E.); ak1686@mynsu.nova.edu (A.K.); aj1262@mynsu.nova.edu (A.J.); mp2063@mynsu.nova.edu (M.P.); 3Dr. J.P. Hussman Institute for Human Genomics, Miller School of Medicine, University of Miami, Miami, FL 33136, USA; dvanbooven@med.miami.edu; 4Department of Veterans Affairs, Miami VA Healthcare System, Geriatric Research Education and Clinical Center (GRECC), Miami, FL 33125, USA; fcollado1@nova.edu; 5South Florida Veterans Affairs Foundation for Research and Education Inc., Fort Lauderdale, FL 33125, USA; 6School Medicine, Universidad Católica de Valencia San Vicente Mártir, 46001 Valencia, Spain; elisa.oltra@ucv.es

**Keywords:** sex differences, ME/CFS, transcriptomics

## Abstract

Myalgic encephalomyelitis/chronic fatigue syndrome (ME/CFS) is a complex, multi-symptom illness characterized by debilitating fatigue and post-exertional malaise (PEM). Numerous studies have reported sex differences at the epidemiological, cellular, and molecular levels between male and female ME/CFS patients. To gain further insight into these sex-dependent changes, we evaluated differential gene expression by RNA-sequencing (RNA-Seq) in 33 ME/CFS patients (20 female, 13 male) and 34 matched healthy controls (20 female and 14 male) before, during, and after an exercise challenge intended to provoke PEM. Our findings revealed that pathways related to immune-cell signaling (including IL-12) and natural killer cell cytotoxicity were activated as a result of exertion in the male ME/CFS cohort, while female ME/CFS patients did not show significant enough changes in gene expression to meet the criteria for the differential expression. Functional analysis during recovery from an exercise challenge showed that male ME/CFS patients had distinct changes in the regulation of specific cytokine signals (including IL-1β). Meanwhile, female ME/CFS patients had significant alterations in gene networks related to cell stress, response to herpes viruses, and NF-κβ signaling. The functional pathways and differentially expressed genes highlighted in this pilot project provide insight into the sex-specific pathophysiology of ME/CFS.

## 1. Introduction

Myalgic encephalomyelitis/chronic fatigue syndrome (ME/CFS) is a complex illness that is estimated to affect up to 2.5 million Americans [1]. Patients with ME/CFS experience unexplained fatigue, post-exertional malaise (PEM), neurological manifestations, autonomic dysfunction, and sleep disturbances [2]. These symptoms are often debilitating, interfering with the ability to perform activities of daily life, with an estimated 25 percent being completely housebound and only 13 percent of ME/CFS patients able to hold full-time employment [1,3]. To date, the etiology of ME/CFS is largely unknown, but is thought to be a combination of environmental exposures, physical/emotional trauma, genetics, and infections (such as human herpesvirus 6 (HHV6), Epstein–Barr Virus (EBV), *Coxiella burnetii*, and others) [1].

There are no specific diagnostic tests for ME/CFS. The diagnosis is based upon history and physical examination findings that meet certain criteria established by the National Academy of Medicine (and other medical bodies), as well as limited laboratory testing to exclude other causes of fatigue [2]. As a result, up to 91 percent of affected patients are misdiagnosed with other conditions such as depression and/or fibromyalgia [1]. In addition, there is no curative therapy for ME/CFS. As such, management is supportive and focuses on treating present symptoms. This unmet need in the diagnosis and treatment of ME/CFS patients is of increasing importance due to the emergence of COVID-19. After getting infected with SARS-CoV-2, a significant percentage of individuals remain ill for many months with a condition similar in presentation to ME/CFS, referred to as Long-COVID syndrome or post-acute sequelae COVID-19 (PASC). A recent study found that six months after a moderate acute COVID-19 illness, half of the patients met the criteria for ME/CFS [4]. As such, advancing our understanding of ME/CFS is critical for the development of diagnostic biomarkers and targeted therapies not only for patients suffering with ME/CFS but also for those with Long-COVID syndrome.

Since ME/CFS was first described in the 1980s, research has shown that the disease is more prevalent in women compared to men. Women also report worse symptom severity, reduced functional capacity, and prolonged flare ups compared to men with the disease [5]. More recently, research efforts have also demonstrated that Long-COVID syndrome has also been associated with female sex and women were more likely to describe symptoms of weakness, fatigue, and palpitations [6].

Sex differences in ME/CFS patients have also been reported at the molecular level. Studies have described differentially expressed genes (DEGs) [7], microRNAs (miRNA) [8], and metabolic profiles [9] between men and women with ME/CFS. These distinct changes have been documented in ME/CFS patients at rest. Meanwhile, the most severe symptomatology presents in response to stress. By subjecting patients to physical stress using an exercise challenge, researchers can further characterize the underlying pathological processes associated with fatigue and PEM. For example, differences in miRNA profiles [10] and metabolomic pathways [11] have been characterized between ME/CFS males and females in response to a physical stressor.

To build upon these insights, our group developed a systematic and reproducible exercise challenge designed to trigger PEM. We then analyzed differences in gene expression in peripheral blood mononuclear cells (PBMC) between the sexes utilizing RNA-sequencing (RNA-Seq) at three different time points: baseline before exercise (T0), maximal exertion (T1), and four hours after maximal exertion (T2). This method allowed us to find evidence of unique immunological changes in response to exertion in ME/CFS females at the molecular level [12]. 

This pilot project was designed to extend our exercise challenge protocol and subsequent analysis to ME/CFS males and sex-matched controls to determine whether our previous findings were sex-dependent. The results indicate that transcriptomic changes differ between females and males with ME/CFS, both in response to exercise, and during recovery (when ME/CFS patients often experience PEM). The pathways and genes highlighted in this pilot project provide novel insights into sex-specific ME/CFS pathogenesis and demand an exploration of sex-based ME/CFS diagnostics and therapeutics.

## 2. Results

### 2.1. Participant Characteristics 

Twenty female and 13 male ME/CFS subjects, and 20 female and 14 male matched healthy controls (HCs) underwent the exercise challenge. No significant differences were observed for age and body mass index (BMI) between individuals with ME/CFS and HCs (Table 1 and [12]). The levels of disability in both physical and mental health were evaluated using the Short Form 36-item Survey (SF-36) questionnaire [13]. Scores were stratified to a 100-point scale, with lower scores representing a higher level of disability. The female ME/CFS cohort had poor (self-reported, *p* < 0.05) outcomes in all reported domains of SF-36 except mental health (*p* = 0.595), as compared to female HCs [12]. Male ME/CFS patients showed a similar pattern in the SF-36 except that no significant difference was found for bodily pain (*p* = 0.37) and mental health (*p* = 0.14) as compared to male HCs (Table 1).

### 2.2. Transcriptomic Changes between Maximal Exertion (T1) and Baseline before Exercise Challenge (T0) Stratified by Sex 

We examined the expression of genes on autosomal chromosomes in PMBCs of both ME/CFS patients, and HC between T1 and T0, and stratified the analysis by the sex of the participants. Differentially expressed genes in females and males were input into Metascape to conduct a functional comparative analysis between the sexes [14]. 

As previously reported, female HCs showed significant changes in the expression of 102 genes, while female ME/CFS patients did not have DEGs that passed the criteria for significance [12]. 

In male HCs, no DEGs passed the criteria for significance (FDR < 0.1). In males with ME/CFS, 118 DEGs showed significant changes in expression: 36 genes were overexpressed, and 72 genes were underexpressed (Appendix A). A functional pathway analysis revealed that gene networks related to the IL-12 pathway, the positive regulation of a response to an external stimulus, and the interactions of natural killer cells were significantly affected in this cohort (Figure 1).

As male HCs did not show any DEGs that passed the criteria for significance between T1 and T0; a functional pathway comparison to female HCs was not performed. 

Female ME/CFS patients also did not show any DEGs that passed the criteria for significance between T1 and T0 [12]; therefore, a functional pathway analysis comparison to male ME/CFS patients was not performed. Therefore, the functional pathways exhibiting significant changes (Figure 1) were unique to ME/CFS males between T1 and T0.

#### Cell Type Abundance Changes (between T1 and T0) Stratified by Sex

The abundances of cell types were estimated using normalized gene counts, as previously described for the female cohort [12]. Unexpectedly, and similar to our results in female HCs [12], male ME/CFS patients had a significant decrease in naive CD4+ T cells and a significant increase in NK cells (Table 2). 

### 2.3. Transcriptomic Changes between 4 h after Maximal Exertion (T2) and Maximal Exertion (T1) Stratified by Sex 

Next, we investigated gene expression in PBMCs from ME/CFS patients and HC between time points T2 and T1, again stratifying by sex. In female HCs, 831 genes had significant changes in their levels of expression with 542 being overexpressed. Female ME/CFS patients had 1277 genes that were differentially expressed, including 892 which were upregulated. The functional analysis of DEGs resulted in several clusters that were significantly affected [12]. 

In male HCs, 1256 DEGs were identified between recovery (T2) and maximal exertion (T1), including 782 that were underexpressed and 474 that were upregulated (Appendix A). The pathway analysis of DEGs resulted in several gene networks that were significantly affected in male HCs as compared to ME/CFS males. These included transcriptional regulation of granulopoiesis, FTL3 signaling, necroptosis, regulation of leukocyte activation, and NK-cell-mediated cytotoxicity. 

Male ME/CFS patients had 1040 genes that were differentially expressed with 535 genes underexpressed, and 505 genes that were overexpressed (Appendix A). Through the pathway analysis, it was discovered that male ME/CFS patients exhibited significant effects on several gene ontologies including inflammatory response, positive regulation of cytokine production, negative regulation of cell differentiation, regulation of IL-4 production, and regulation of inflammatory response.

#### 2.3.1. Cell Type Abundance Changes (between T2 and T1)

In female HCs, there was a significant decrease in memory B cells, CD8+ T cells, NK cells and eosinophils. In addition, female HCs had significant increases in naive and memory CD4+ T cells, as well as activated mast cells. Female ME/CFS patients showed significant elevations in naive CD4+ T cells while dendritic cells and eosinophils were significantly reduced, as previously described [12]. 

In contrast, male HCs showed significant increases in CD8+ T cells, NK cells, dendritic cells, and eosinophils while naive CD4+ T cells were significantly decreased. Male ME/CFS patients had significant reductions in NK cells, dendritic cells, and eosinophils. In addition, and similarly to ME/CFS females, naive CD4+ T cells were significantly elevated in ME/CFS males (Table 3). 

#### 2.3.2. Male HC vs. Female HC Functional Pathway Analysis (between T2 and T1)

To identify sex differences, we analyzed the DEGs of male and female HC participants during their recovery from an exercise challenge and conducted a comparative functional analysis of the gene networks. In female HCs, the most-impacted gene ontologies were regulation of defense response, mononuclear cell proliferation, regulation of protein transport, immune effector process, and positive regulation of migration (Figure 2). Meanwhile, in male HCs the most affected gene networks included transcriptional regulation of granulopoiesis, herpes simplex virus 1 infection, response to peptide, response of molecule to bacterial origin, and signaling by Rho GTPases (Figure 2).

#### 2.3.3. Male ME/CFS vs. Female ME/CFS Functional Pathway Analysis (between T2 and T1)

To identify sex differences during the onset of PEM, we did a comparative functional analysis of DEGs from ME/CFS females and males during recovery from exercise. Gene ontologies that were the most affected in female ME/CFS as compared with male ME/CFS patients included herpes simplex virus 1 infection, regulation of cellular response to stress, cellular responses to stress, positive regulation of cell death, and membrane organization (Figure 3). Pathways that were the most affected in male ME/CFS as compared to female ME/CFS patients included regulation of interleukin-1 beta production, regulation of nervous system development, negative regulation of cell development, inflammatory response, and cell activation (Figure 3).

### 2.4. Transcriptomic Changes between 4 h after Maximal Exertion (T2) and Baseline (T0) Stratified by Sex

Finally, we analyzed gene expression in PBMCs from ME/CFS patients and HCs between time points T2 and T0, stratifying by sex. Female HCs had 592 genes that showed significant changes in their levels of expression, with 462 being downregulated and 100 being upregulated. The pathway analysis of the DEGs showed that gene ontologies related to IL-18 signaling, gland development, tube morphogenesis, Epstein–Barr virus infection and NF-kappa signaling were the most significantly affected. In female ME/CFS, 1059 were differentially expressed, including 740 that were overexpressed and 319 that were underexpressed. The functional analysis of DEGs showed that gene networks associated with the regulation of cell activation, herpes simplex virus 1 infection, cytokine signaling in the immune system, regulation of leukocyte apoptotic processes and positive regulation of cell death showed the most significant alterations. 

In male HCs, 1048 DEGs identified between recovery (T2) and baseline (T0), including 612 genes that were underexpressed and 436 that were overexpressed (Appendix A). Gene networks that were the most significantly changed included endocytosis, brain development, positive regulation of catabolic process, autophagy, and regulation of vesicle-mediated transport. Male ME/CFS patients had 1516 genes that were differentially expressed with 459 being upregulated and 1057 being downregulated (Appendix A). The functional analysis of the DEGs showed enrichment of pathways related to formation of the beta-catenin: TCF transactivating complex, viral-infection pathways, herpes simplex virus 1 infection, cytokine signaling in the immune system and H2AX complex II.

#### 2.4.1. Cell Type Abundance Changes (between T2 and T0)

Female HCs, female ME/CFS patients and male HCs showed significant increases in activated dendritic cells and eosinophils as well as a noteworthy increase in naive CD4+ T cells when comparing baseline cell type abundance to 4-h post recovery. In comparison, male ME/CFS patients had an increase in memory CD4+ T cells (Table 4).

#### 2.4.2. Male HC vs. Female HC Functional Pathway Analysis (between T2 and T0)

To elucidate sex-specific differences between healthy control cohorts, we analyzed DEGs at baseline, and four hours after the exercise challenge between female and male HCs and performed a comparative functional analysis. The gene ontologies that were more significantly affected in female HCs included the IL-18 signaling pathway, response to wounding, formation of snRNP, regulation of apoptotic signaling pathway and NF-kappa B signaling pathway (Figure 4). Meanwhile, male HCs showed significant changes in pathways related to herpes simplex virus 1 infection, H2AX complexes, cellular senescence, exercise-induced circadian regulation, and negative regulation of the immune system (Figure 4).

#### 2.4.3. Male ME/CFS vs. Female ME/CFS Functional Pathway Analysis (between T2 and T0)

In order to discern sex-specific differences between ME/CFS patients, we extracted DEGs from female and male ME/CFS patients during recovery from exercise and at baseline and conducted a comparative functional analysis. Gene networks that were the most affected in female ME/CFS patients as compared to males ME/CFS patients included cytokine signaling in the immune system, cellular response to stress, hematopoiesis, positive regulation of cell death and negative regulation of immune system processes (Figure 5). Conversely, gene ontologies that were the most affected in males with ME/CFS included PDGFRB, brain development, membrane organization, adaptive immune response, and cellular response to DNA-damage stimulus (Figure 5).

### 2.5. Nanostring Validation

Approximately 90 percent of the DEGs identified by RNA-Seq were validated by Nanostring technology (Appendix A). Genes with discrepancies between Nanostring and RNA-seq were not used in the functional analysis.

## 3. Discussion

This pilot project is the first exercise challenge study in ME/CFS patients that evaluated sex differences utilizing RNA-Seq. The results have highlighted that biologic sex influences differential gene expression of PBMCs in response to exercise and during recovery. 

### 3.1. Transcriptomic Changes between Maximal Exertion (T1) and Baseline before Exercise (T0) in ME/CFS Patients and HCs 

Overall, male ME/CFS patients and female HCs both had significant changes in gene expression in response to an exercise challenge. Meanwhile, male HCs and female ME/CFS patients did not show significant enough transcriptomic changes to meet the criteria for this pilot analysis (FC > 1.5 in either direction, FDR < 0.1). The finding of no change in lymphocyte gene expression upon reaching maximal exertion in male HCs is not fully understood at this time. 

Males with ME/CFS had upregulation of genes related to receptors for immunoglobulins, chemokines, and other signaling molecules of the immune system, including *KRI3DL2*, *CD160* and *CCL4* (Appendix A) in response to exertion. Genes that were downregulated included those involved in the transcription of DNA into RNA, including *LINCOO342*, *POLR2J4*, *PRR12* (Appendix A). Functional analysis of these gene networks showed changes in pathways related to the IL-12 pathway, positive regulation of response to an external stimulus, and NK cell function. 

The functional analysis of DEGs in female HCs showed that the most affected pathways were NK-cell-mediated cytotoxicity, immunoregulatory actions, leukocyte activation, and the IL-12 pathway. Despite an overlap of the functional response in males with ME/CFS to female HCs—for example, derangement of the IL-12 pathway—the genes most affected in each group were different, with female HCs having significant alterations in multiple DEGs related to lymphocyte differentiation, signaling, and fate, including *MAL* and *SLC16A10* [12].

Male ME/CFS patients had the most significant transcriptomic changes in the IL-12 pathway. IL-12 is known to stimulate interferon and tumor necrosis factor (TNF) production by NK cells and T cells [16]. Several studies have demonstrated that circulating IL-12 levels are elevated in ME/CFS patients as compared to HCs [17,18,19]. Increased levels of proinflammatory cytokines have been associated with fatigue [20]. In addition, IL-12 has been associated with reduced measures of T-cell metabolism in ME/CFS patients [21]. Dysregulation of IL-12 production in males with ME/CFS may cause dysfunction in downstream cytokine signaling and altered energy production in immune cells leading to PEM in response to exercise [22]. Changes seen in this pathway may partly explain why men with ME/CFS do not experience symptoms in response to exertion as severe as those of female ME/CFS patients [5]. Female HCs (like male ME/CFS patients) had significant changes in the IL-12 pathway in response to the exercise challenge, but female ME/CFS patients did not. These similar changes in the IL-12 pathway between male ME/CFS patients and HCs may allow for the reduction of symptoms associated with cytokines, including fatigue and pain. To further illustrate this point, male ME/CFS patients had similar levels of bodily pain, as measured by the SF-36 questionnaire, as compared to male HCs, while female ME/CFS patients had significantly more bodily pain as compared to female HCs (Table 1 and [12]). 

In addition, recent research has highlighted the role of estrogen receptors in the regulation of IL-12 production [23]. The two main subtypes of estrogen receptors (ER) have different effects on IL-12 expression. For example, ERα has been shown to positively regulate IL-12 production, while ERβ has been found to have a negative regulatory effect. Therefore, if male ME/CFS patients and female HCs are able to efficiently regulate IL-12 production, this may suggest dysregulation of estrogen receptors in female ME/CFS patients as a potential driving force behind the observed difference in cytokine function.

Overall, the role of IL-12 in ME/CFS is still unclear. Further exploration of this topic is warranted as this cytokine and its downstream signaling may either induce symptoms of PEM in males with ME/CFS, or it could play a part in why males with ME/CFS experience less severe PEM than their female counterparts.

ME/CFS male patients had significant changes in the functional pathways responsible for positive regulation of a response to an external stimulus. This is a broad pathway that encompasses any process that activates, maintains, or increases the rate or cellular response to any external stimulus [24]. The data suggests that males with ME/CFS activated a cellular response to a physical stressor, whereas female ME/CFS patients did not. These molecular reactions may be detrimental to male ME/CFS patients. For example, metabolomic studies have found deficits in pathways that generate energy from sugars, fatty acids, and amino acids in ME/CFS patients [25,26,27]. At the present time, it is unclear whether activating a cellular response contributes to the pathophysiology of ME/CFS in males, or whether it is part of the reason why males exhibit less severe symptoms than females. 

Males with ME/CFS had significantly affected NK cell function compared to male HC. NK cells are cytotoxic immune cells of the innate immune system that function to lyse foreign invading pathogens and somatic cells with internal damage [28]. NK-cell cytotoxicity and aberrant lytic protein levels are altered in ME/CFS patients [29]. As such, these findings suggest that males with ME/CFS are activating pathways in dysfunctional NK cells. This dysregulated innate immune function may contribute to the symptoms of PEM in males with ME/CFS. The described increases in KIR3DL1 gene expression and cytotoxic-related cytokines is suggestive of impairments in the NK-cell cytotoxic pathways, in particular, the granule-dependent and -independent pathways. Our data indicate that males with ME/CFS show similar changes in the functional pathways related to NK cells as female HCs (Figure 1 and [12]). In addition, both male ME/CFS patients and female HCs had significant elevations in the levels of NK cells as measured by cell type abundance (Table 2 and [12]). This warrants further investigation to understand if abnormal cytotoxic profiles of NK cells are found in female ME/CFS patients only, or in both sexes, which can therefore be a focus for future diagnostic parameters and targeted therapy. 

The changes in gene expression and functional pathways in response to exercise (between T1 and T0) seen in male ME/CFS patients in this pilot project may differentiate them from male HCs and females with ME/CFS. These pathways warrant exploration for diagnostic biomarkers that correspond to the progression of the disease in male patients with ME/CFS. In addition, the transcriptomic alterations seen only in female HCs highlight signaling molecules and immune cells that may be worth further investigation to characterize their role in ME/CFS female pathophysiology. Overall, it is largely unclear whether the observed sex differences are contributing to ME/CFS symptoms, or whether they explain the symptomatic differences between males and females with ME/CFS. 

### 3.2. Transcriptomic Changes between 4 h after Maximal Exertion (T2) and Maximal Exertion (T1) in ME/CFS Patients and HCs 

When comparing timepoints T2 to T1, all four cohorts had significant changes in gene expression. Male ME/CFS patients downregulated genes that are involved in the metabolic reprogramming of immune cells (such as SLC7A5 and CREM). Meanwhile genes related to immune cell infiltration and survival like GIMAP8, GIMAP5 and GIMAP4 were upregulated (Appendix A). In females with ME/CFS, genes related to cytokine signaling were overexpressed, while those related to helping protect cells from the adverse effects of stress were downregulated [12]. 

Between T2 and T1, female HCs had significant downregulation of genes related to inflammatory signaling in the lymphocytes [12]. Male HCs had upregulation of genes related to cytokine signaling including CD180, CCR2 and TLR10. They also showed downregulation of genes related to cellular stress, division and apoptosis including DDIT3, PLK2 and MAFF (Appendix A). 

It is important to highlight the most significant pathway differences between male and female HCs to provide insight into how healthy individuals regulate transcription in response to exercise. Male HCs, compared to female HCs, had the most significant changes in transcriptional regulation of granulopoiesis. Granulopoiesis is responsible for the formation of neutrophils, basophils, and eosinophils which play a role in mediating the inflammatory process [30]. This difference was also reflected in cell type abundance data showing that male HCs had significant increases in levels of eosinophils while female HCs did not (Table 3 and [12]). Meanwhile, female HCs had the most significant changes in the pathways related to mononuclear cell proliferation (Figure 2C). Mononuclear cells include any blood cell with a round nucleus (i.e., lymphocytes, monocytes, NK cells or dendritic cells) [31]. This underlying variation in PBMC development between male and female HCs in response to recovery from exercise may be exacerbated in patients with ME/CFS and may be a potential avenue for further research into predisposition to the disease. 

Female ME/CFS patients showed more significant alterations in response to Herpes simplex viruses as compared to males. This data suggests that the immune system of female ME/CFS patients interacts with herpes viruses in a different way from male ME/CFS patients. The majority of ME/CFS cases begin after exposure to viral infection, especially those from the herpesvirus family [32]. Herpes viruses are known for their ability to establish lifelong infections, taking up residency in nerve cells [33]. Immune cell exhaustion has been observed in mice, where the chronically persistent virus rendered certain subsets of lymphocytes unable to kill virally infected cells [34]. It is possible that a similar phenomenon is occurring here in females with ME/CFS, specifically during the recovery period after exercise. These transcriptomic changes may point to a common infectious trigger for ME/CFS in females. As such, further exploration of this topic may be worthy of exploration as a contributor to the female predisposition to ME/CFS. 

Females with ME/CFS had significantly changed cellular responses to stress compared with males with ME/CFS. These pathways are essential for maintaining homeostasis in response to an external stressor. In this pilot project, we used exercise as a model of stress. Stress can increase the production of free radicals. To combat these molecules, the body relies on antioxidant defenses. Studies have shown that ME/CFS patients have reduced levels of antioxidants and impaired redox status which was correlated with the severity of patients’ symptoms [35]. Another study showed increased levels of oxidative and nitrosative stress in ME/CFS patients, including increased levels of isoprostanes, peroxides, and superoxide [36]. Together, these findings suggest that females with ME/CFS may be more vulnerable to cell death after significant exertion due to their inability to properly defend against oxidative and nitrosative stress. 

We found that there were significant transcriptomic alterations in NF-κB signaling in females with ME/CFS compared ME/CFS males. NF-κB induces the expression of various pro-inflammatory genes, including those encoding cytokines and chemokines, and participates in inflammasome regulation [37]. Dysregulation of NF-κB contributes to the pathogenic process of various inflammatory diseases [38]. More importantly, increased levels of NF-κB proteins have been shown in the PBMCs of ME/CFS patients [39]. ME/CFS transcriptomic studies have also found upregulation of genes related to a biological-counter response to unwanted excess activity of NF-κB [40]. This data suggests that significant transcriptomic changes in NF-κB may be contributing to the inflammatory processes relating to PEM in ME/CFS females. 

Chronic inflammation and increased production of various proinflammatory cytokines are common in ME/CFS patients [18,41,42]. This is particularly relevant as the level of proinflammatory cytokines correlates with fatigue in ME/CFS. Male and female ME/CFS patients both showed changes in pathways related to cytokine signaling. Female ME/CFS patients had functional changes in pathways such as intracellular protein transport and membrane organization pathways that occur due to increases in cytokine signaling (Figure 3B,C). On the other hand, male ME/CFS patients had significant changes in regulation of the IL-1β pathway (Figure 3B,C). Increases in IL-1β have been reported higher in patients with ME/CFS than in HC [43]. A recent study found that IL-1β is one of the most discriminatory cytokines for PEM in patients with ME/CFS [43,44,45]. Regulation of IL-1β pathways is more significantly affected in males with ME/CFS compared to females during recovery from exercise (Figure 3C). This marker may be worthy of exploration as a diagnostic biomarker for disease development and progression, especially in male patients with ME/CFS. 

Overall, during the recovery period from exercise as compared to maximal exertion, there were significant differences observed between males and females with ME/CFS relating to herpes simplex virus 1, inflammatory pathways related to cytokine signaling, and NF-κB signaling. 

### 3.3. Transcriptomic Changes between 4 h after Maximal Exertion (T2) and Maximal Exertion (T0) in ME/CFS Patients and HCs 

We compared gene expression changes 4 h after the exercise challenge to baseline gene expression. Interestingly, both female and male ME/CFS patients showed similar upregulation of genes related to receptors and proteins that are essential for the development, homeostasis, and survival of immune cells (CXCR2R1, CD200R, CD180, and GIMAPs) and downregulation of genes related to the production of interferons (SCD2 and DDIT3) and metabolism in immune cells (SLCs). Male HCs had a decreased expression of genes responsible for T cell activation, differentiation, and function (NRP4A3) and inflammasome formation (NRIP3). On the other hand, female HCs reduced the expression of genes associated with T cell differentiation and cytokine receptors and augmented genes associated with naive T cells, long non-coding RNAs, and chemokines.

To gain a better understanding of how sex differences impact transcriptional regulation of the response to recovery from exercise, we performed a function comparison of DEGs between male and female HCs at timepoints T2 and T0 (Figure 4 and Appendix A). Male HCs exhibited significant changes in pathways related to adapting to the stress caused by exercise, such as by enriching pathways related to repairing cellular damage, including H2AX II complex and cellular senescence. They also showed substantial alterations in pathways involved in the adaptive immune system and negative regulation of immune system processes. These findings highlight that exercise has dynamic effects on immune-cell function mediated by many different cytokines, hormones, and gene expression changes [46]. Similarly, female HCs showed notable changes in pathways related to restoring cellular integrity after exercise, including apoptosis and wound healing. Female HCs also modulated their immune system through changes in cytokines, such as IL-18 and NF-kappa β signaling (Figure 4C). Compared to baseline, after an exercise challenge, HCs exhibited parallel changes in pathways aimed at adapting to exercise by repairing or removing damaged cells and modifying immune-cell parameters. 

In stark contrast, male and female ME/CFS patients showed significant differences in enriched pathways when comparing baseline gene expression to changes in gene expression 4 h post-exercise challenge (Figure 5 and Appendix A). For example, male ME/CFS patients had the most significant changes in the PDGFRB pathway (Figure 5C). Studies have shown that PDGFs can inhibit natural killer cell cytotoxicity when added to a mixed lymphocyte population [47], and their downstream effects are potent modifiers of T-cell lymphokine production [48]. Furthermore, research has demonstrated that PDGF-null mice alter the pattern of macrophage and lymphocyte gene expression to a more proinflammatory phenotype, and these changes are associated with an increase in the percentage of activated T cells [49]. The considerable enrichment of the PDGF pathway may suggest that male ME/CFS patients have dysregulated PDGF, as ME/CFS is often associated with chronic inflammation. Additionally, PDGFs are key master regulators in the survival pathways of many different cancers [50] and may serve to preserve the lifespan of dysfunctional inflammatory cells. Further research is needed to understand the role of this growth factor in the pathophysiology of ME/CFS and how it contributes to the development of PEM in male ME/CFS patients. 

Compared to male ME/CFS patients, female ME/CFS patients had the most significant alterations in pathways related to cytokine signaling and cellular-stress responses (Figure 5). We found distinct changes in IL-10 signaling among ME/CFS females. IL-10 is a cytokine known for its anti-inflammatory properties and its role in regulating immune responses [51]. Some studies have reported higher levels of IL-10 in ME/CFS patients’ blood [52]. The paradoxical enrichment of the IL-10 pathway in our study may suggest resistance to IL-10′s anti-inflammatory actions in ME/CFS females, as was shown in individuals with Type 2 diabetes [53]. From the other side, IL-10 has been shown to act as a pro-inflammatory and immunostimulatory molecule under certain contexts [54]. Interestingly, several studies have predicted poor outcomes in patients with COVID-19, suggesting that dysregulation of IL-10 can occur during states of excessive immune cell activation, such as the cytokine storm seen with SARS-CoV-2 infection [55]. 

We also found that female ME/CFS patients had unique changes in the ferroptosis pathways (Figure 5C). Ferroptosis is an iron-dependent, non-apoptotic mode of cell death that is characterized by the accumulation of lipid reactive oxygen species [56]. Cells undergoing ferroptosis show reduced mitochondrial volume, increased bilayer density and the reduction or disappearance of mitochondrial cristae [57]. Mitochondrial dysfunction has been implicated in the pathophysiology of ME/CFS [58]. Therefore, due to mitochondria malfunction, ME/CFS females may have dysregulated ferroptosis or be acutely predisposed to this process after a stressor. In addition, ferroptosis mainly involves changes in iron homeostasis and lipid-peroxidation metabolism. Several recent studies have found that patients with ME/CFS have increased levels of lipid-peroxidation markers [59]. Taken together, these findings suggest that ferroptosis may play a role in the pathogenesis of PEM in female ME/CFS patients. In addition, the heart highly depends on functional mitochondria. ME/CFS research should be correlated with maladaptive physiological changes that occur during, and after, exercise. For example, dysfunctional right ventricular remodeling during exercise may overlap with those of a very early form of arrhythmogenic right-ventricular cardiomyopathy [60]. Further research is needed to fully understand the link between ferroptosis, mitochondrial dysfunction and ME/CFS, as well as to determine if targeting ferroptosis could be a potential therapeutic strategy for women suffering from this debilitating condition. 

## 4. Materials and Methods

### 4.1. Cohort

This pilot project was conducted in the Miami/Fort-Lauderdale area and included 24 female subjects clinically diagnosed with ME/CFS and 21 female HCs, as well as 11 male ME/CFS subjects and 14 male HCs. All individuals with ME/CFS and HCs were recruited as a part of a large ongoing study by Nova Southeastern University (NSU) Dr. Kiran C. Patel College of Osteopathic Medicine (KPCOM) Institute for Neuro-Immune Medicine. The study was conducted in accordance with the Declaration of Helsinki. The protocol was approved by the institutional review board of NSU (2016-2-NSU) and all subjects provided informed consent. ME/CFS subjects met specific criteria for diagnosis (including the 1996 Center for Disease Control and Prevention (CDC)/ Fukuda and 2003 Canadian Case definitions for ME/CFS). Patients were excluded if they had a history of active smoking or alcohol, diabetes, immunodeficiency, cardiovascular disease, stroke, autoimmunity, malignancy, or systemic infection within two weeks of blood collection. The two cohorts were matched for age and body mass index. The SF-36 questionnaire was used to evaluate physical and mental health between ME/CFS patients and HCs. Scores on this questionnaire range from 0 to 100 and higher scores indicate less disability. Female subjects also completed a gynecologic questionnaire to ensure that blood collection occurred during the first two weeks of their menstrual cycle. The SF-36 questionnaire was used to compare individuals with ME/CFS and HCs including “Physical Health” (which is comprised of eight domains of well-being including physical functioning, physical role functioning, bodily pain, general health perception) and “Mental Health” (which includes, vitality, social functioning, emotional role, and mental health). Each of these eight domains is transformed into a zero-to-100-point scale, with each question being weighed equally. Higher scores on this scale indicate lower levels of disability. In other words, a score of zero is demonstrative of maximal disability and a score of 100 is demonstrative of no disability in that particular domain [13]. The SF-36 consists of eight scaled scores, which are the weighted sums of the questions in their section. Each scale is directly transformed into a zero to 100 scale on the assumption that each question carries equal weight. The lower the score, the greater the disability. The higher the score, the greater the disability (i.e., a score of zero is equivalent to maximum disability and a score of 100 is equivalent to no disability). Female subjects also completed a gynecologic questionnaire to ensure blood collection occurred during the first two weeks of their menstrual cycle. 

Participants ate a uniform breakfast (yogurt and banana) and rested for 30 min in reclining chairs before blood was drawn (T0). Subsequently, a standard maximal graded exercise test (GXT) was conducted using McArdle’s protocol [61]. This protocol was used as part of a larger ongoing study to investigate the biological mechanisms underlying neuroimmune diseases. The GXT involved participants pedaling at 60 W for two minutes, with an increase of 30 W every two minutes until they reached their maximum exertion. The second blood draw was taken at the point of maximal exertion (T1) and the third blood draw was collected four hours after maximal exertion (T2). 

### 4.2. PBMC Isolation and RNA Extraction

Each participant provided up to 8 mL of whole blood, which was collected in K2EDTA tubes and diluted at 1:1 (*v*/*v*) ratio in phosphate-buffered saline (PBS) within two hours. The resulting solution was then layered on top of Ficoll-Paque Premium (GE Healthcare, Chicago, IL, USA) and subjected to density centrifugation at 500× *g* for 30 min at 20 °C with the brakes off. This process facilitated the isolation of the PBMC layer, which was then washed with PBS, resuspended in one volume of red blood cell lysis buffer, kept on ice for five minutes, and subsequently centrifuged at 500× *g* for ten minutes. Finally, the PBMC pellets were resuspended in a freezing medium and aliquots of 10^7^ cells/mL were frozen in liquid nitrogen until required for analysis. Total RNA was extracted using RNAzol (Molecular Research Center, Cincinnati, OH, USA), and the quality of RNA was assessed using Agilent TapeStation 4200, with all RNA samples having an RNA integrity number greater than seven.

### 4.3. RNA Sequencing

A total of 500 ng of RNA was submitted to the Center of Genome Technology (CGT) at the University of Miami and to the Genomic Core Facility of Nova Southeastern University for RNA-Seq. The TruSeq Stranded Total RNA library Prep Kit (Illumina Inc., San Diego, CA, USA) was used to generate libraries with a paired-end sequencing reading length of 150 nucleotides. The Illumina RNA-Seq pipeline was employed to evaluate genomic coverage, percent alignment, and nucleotide quality for quality control assessment. Following quality control, the raw sequencing data were converted to the fastq format.

### 4.4. RNA-Seq Analysis

GSNAP (version 2021-02-22) [62], HISAT2 (version 2.1.0) [63] and STAR (version 2.7.8a) [64] software were employed to map raw reads to the reference human genome (GRCh38, release 86). Counts of reads aligned by GSNAP and HISAT2 were calculated using HTSeq software (version 0.13.5) [65], while STAR alignment was executed with the “--quantMode Transcriptome SAM” option. Following counting by HTSeq and STAR, genes from sex chromosomes were removed, and ComBat-seq [66] was employed for batch correction. The counts were then imported into Bioconductor/R (version 4.2.1) package DESeq2 (version 1.36.0) [67] for differential gene expression analysis. Genes with raw counts of 50 or higher in all samples in at least one group (ME/CFS patients at T0, ME/CFS patients at T1, ME/CFS patients at T2, HC at T0, HC at T1, HC at T2) were utilized for the differential analysis by DESeq2, resulting in a total of 10,120 transcripts. DEGs were selected based on two criteria: (1) either a fold change (FC) > 1.5 and false discovery rate (FDR) < 0.10 in one out of three aligners, and FC > 1.4 and FDR < 0.15 in the other two aligners; or (2) an FC > 1.5 and FDR < 0.10 in at least two out of three aligners. 

After selection, the DEGs were uploaded to Metascape [14], where an Express Analysis was conducted using default parameters, including a *p*-value threshold of <0.01, a minimum count of three, and an enrichment factor of >1.5. Cytoscape [68] was utilized to visualize networks.

We employed CIBERSORTx (https://cibersortx.stanford.edu/, accessed on 2 January 2023) [15] software to examine potential variations in lymphocyte subtypes using normalized gene counts. The default LM22 leukocyte gene signature matrix [15] was employed as a reference. Quantile normalization was deactivated, and 1000 permutations were used. To identify statistically significant differences in cell counts, we used analysis of variance (ANOVA) and established a cutoff *p*-value of 0.05. 

### 4.5. Validation of RNA-Seq Results

A custom panel was utilized on the Nanostring nCounter platform to confirm the RNA-Seq findings. The processing and hybridization of 100 ng total RNA were carried out, followed by counting in accordance with the manufacturer’s guidelines. The raw counts were analyzed using the NanoString nSolver v.4 software. For all samples, the geometric means of negative controls plus two standard deviations were computed, and any count below this threshold value was eliminated from normalization and analysis. Subsequently, all procedures were executed following the manufacturer’s instructions.

## 5. Conclusions

In conclusion, our results clearly delineate differences between the effects of exercise on male and female ME/CFS patients’ transcriptomics as well as during the recovery period when PEM sets in. While this study has yielded valuable insights into the effect of sex on the pathophysiology of ME/CFS, it is important to acknowledge certain limitations. In future studies, larger-scale cohorts are needed to comprehensively examine transcriptomic profiles and identify potential biomarkers associated with disease progression at various time points during and after an exercise challenge, including time points of 48 h and longer. Additionally, integrating other omics data for the same participants at the same time points would provide a more holistic understanding of the molecular mechanisms and pathways involved. Another constraint that must be addressed in future ME/CSF research involving exercise challenges is how to measure PEM. PEM is inherently a subjective experience; thus, this makes it difficult to assess [69,70]. To ensure accurate and consistent assessment of PEM, it is imperative to develop standardized assays that can capture the multidimensional aspects of PEM. It is warranted to include these PEM measures in future “omics” studies of ME/CFS using an exercise challenge. The correlation of transcriptomic and other “omics” changes with measures of PEM at various time points during and after exercise challenge would help to uncover the underlying mechanisms of ME/CFS. 

## Figures and Tables

**Figure 1 ijms-24-10255-f001:**
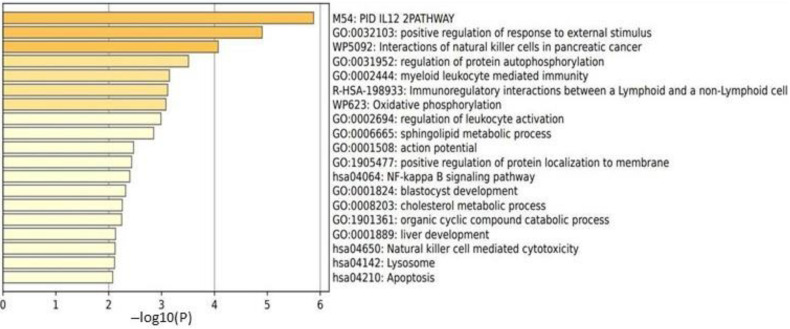
Metascape [14] express analysis of all 108 DEGs in male ME/CFS patients between T1 and T0. Cutoff values included a *p*-value <0.01, a minimum count of 3, and an enrichment factor >1.5. GO biological processes, KEGG pathways, Reactome gene sets, CORUM complexes, and canonical pathways from MSigDB were included in the search.

**Figure 2 ijms-24-10255-f002:**
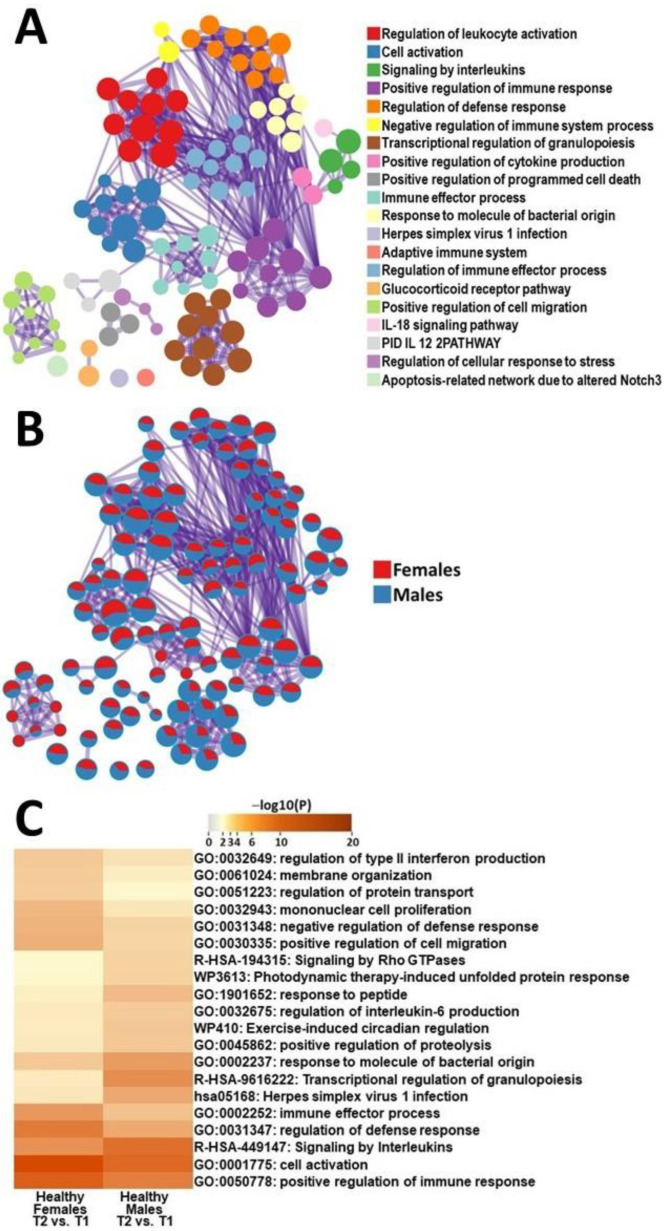
Metascape express [14] analysis of differentially expressed genes between male HCs and female HCs between T1 and T2: (**A**) each node represents an enriched term. The larger the representation of the color the more differentially expressed in each cohort. Cutoff values included a *p*-value of <0.01, a minimum count of 3, and an enrichment factor of >1.5. GO biological processes, KEGG pathways, Reactome gene sets, CORUM complexes, and canonical pathways from MsigDB were included in the search; (**B**) nodes are colored by ratios between male HCs and female HCs. The red color indicates functional pathways in female HCs, while the blue color indicates functional pathways in male HCs; and (**C**) heatmap of the individual pathways that showed the largest difference between T2 and T1 in PBMCs of male HCs and female HCs.

**Figure 3 ijms-24-10255-f003:**
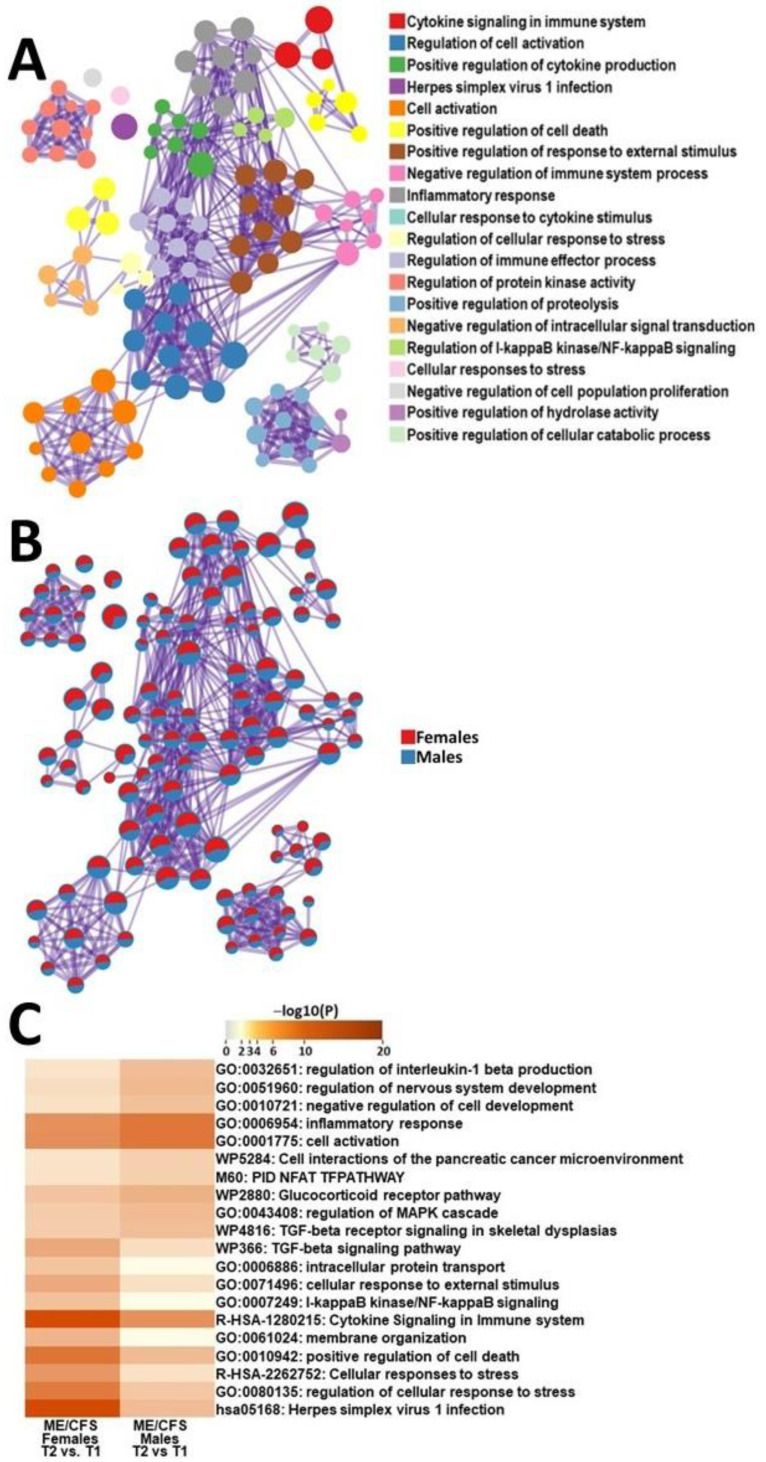
Metascape express [14] analysis of differentially expressed genes between male ME/CFS patients and female ME/CFS patients between T1 and T2: (**A**) each node represents an enriched term. The larger the representation of the color the more differentially expressed in each cohort. Cutoff values included a *p*-value of <0.01, a minimum count of 3, and an enrichment factor of >1.5. GO biological processes, KEGG pathways, Reactome gene sets, CORUM complexes, and canonical pathways from MsigDB were included in the search; (**B**) nodes are colored by ratios between male HCs and female HCs. The red color indicates functional pathways in female ME/CFS patients while the blue color indicates functional pathways in male ME/CFS patients; and (**C**) heatmap of enriched terms across lists of DEGs between T2 and T1 in PBMCs of male ME/CFS patients and female ME/CFS patients.

**Figure 4 ijms-24-10255-f004:**
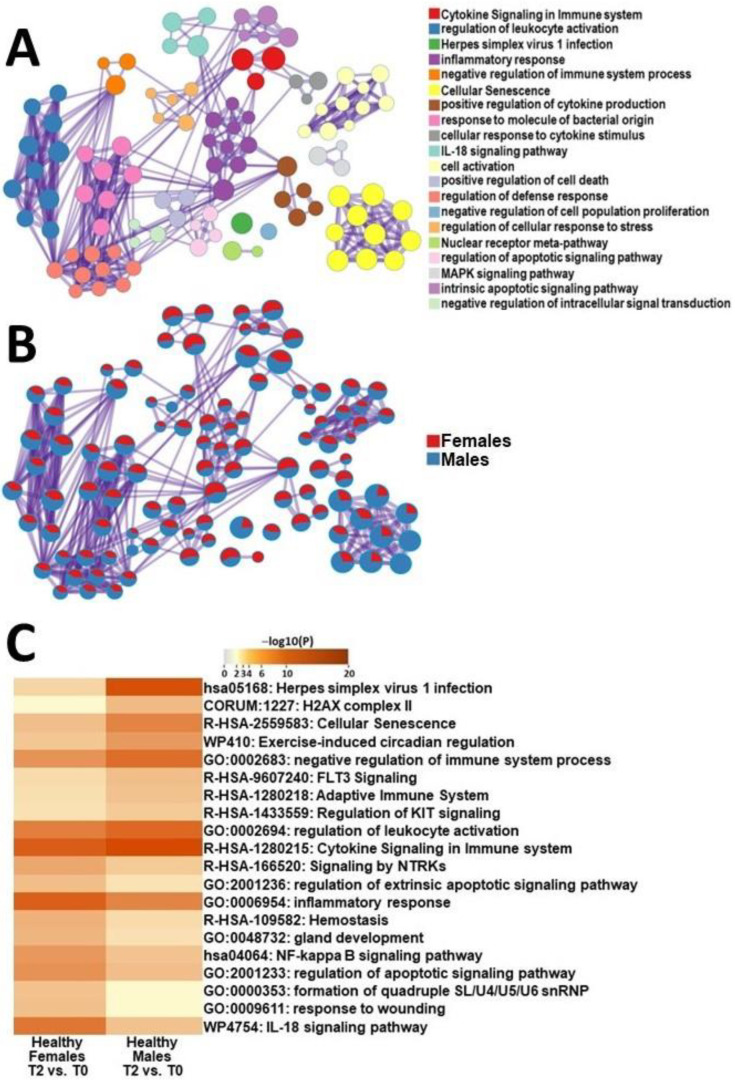
Metascape express [14] analysis of differentially expressed genes between male HCs and female HCs between T2 and T0: (**A**) each node represents an enriched term. The larger the representation of the color the more differentially expressed in each cohort. Cutoff values included a *p*-value of <0.01, a minimum count of 3, and an enrichment factor of >1.5. GO biological processes, KEGG pathways, Reactome gene sets, CORUM complexes, and canonical pathways from MsigDB were included in the search; (**B**) nodes are colored by ratios between male HCs and female HCs. The red color indicates functional pathways in female HCs while the blue color indicates functional pathways in male HCs; and (**C**) heatmap of the individual pathways that showed the largest difference between T2 and T0 in PBMCs of male HCs and female HCs.

**Figure 5 ijms-24-10255-f005:**
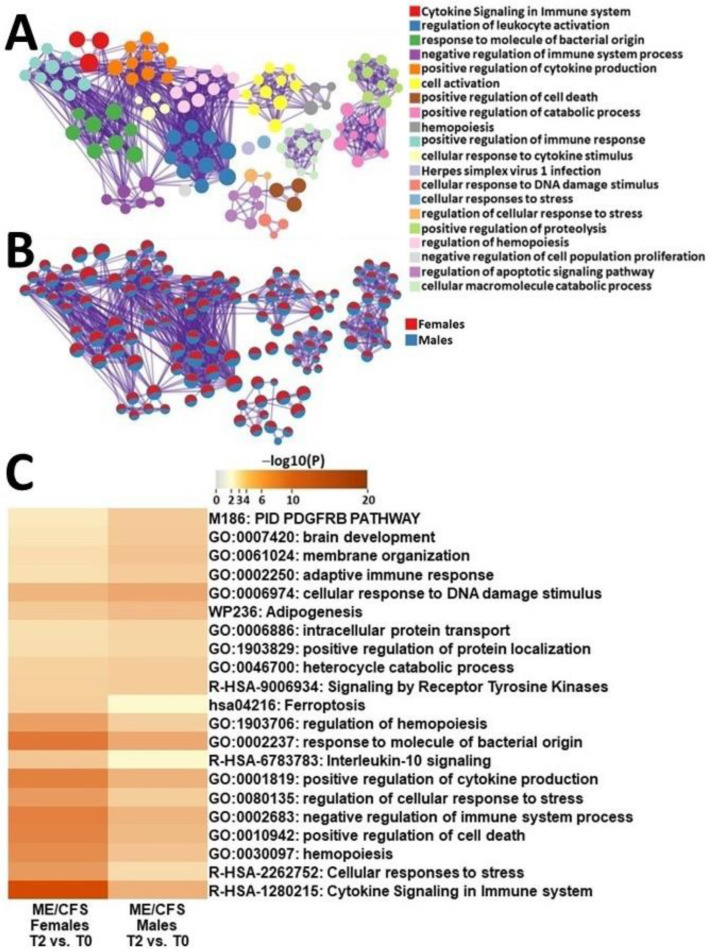
Metascape express [14] analysis of differentially expressed genes between male ME/CFS patients and female ME/CFS patients between T2 and T0: (**A**) each node represents an enriched term. The larger the representation of the color the more differentially expressed in each cohort. Cutoff values included a *p*-value of <0.01, a minimum count of 3, and an enrichment factor of >1.5. GO biological processes, KEGG pathways, Reactome gene sets, CORUM complexes, and canonical pathways from MsigDB were included in the search; (**B**) nodes are colored by ratios between male ME/CFS and female ME/CFS patients. The red color indicates functional pathways in female ME/CFS patients while the blue color indicates functional pathways in male ME/CFS patients; and (**C**) heatmap of enriched terms across lists of DEGs between T2 and T0 in PBMCs of male ME/CFS patients and female ME/CFS patients.

**Table 1 ijms-24-10255-t001:** Demographic information comparing male ME/CFS and male HC including SF-36 questionnaire data. Data are shown as mean ± standard error of mean, *—*p* ≤ 0.05, Student *t*-test.

	Category	ME/CFS Males	Healthy Controls	*p*-Value
	Age	42.2 ± 4.25	45.3 ± 2.67	0.550
	BMI	25.8 ± 1.37	28.8 ± 0.89	0.115
Physical Health				
	Physical Function	52.9 ± 9.03	76.8 ± 7.73	0.050 *
	Role-Physical	16.7 ± 11.24	91.1 ± 5.63	<0.001 *
	Body pain	65.2 ± 6.93	74.1 ± 6.62	0.371
	General Health	34.6 ± 7.16	73.1 ± 5.88	<0.001 *
Mental Health				
	Vitality	26.7 ± 7.24	65.7 ± 6.09	<0.001 *
	Social Function	36.5 ± 7.29	84.8 ± 4.76	<0.001 *
	Role Emotional	63.9 ± 12.62	92.8 ± 5.18	0.050 *
	Mental Health	71.7 ± 5.54	83.1 ± 4.86	0.135

**Table 2 ijms-24-10255-t002:** Cell type abundance from CIBERSORTx [15] analysis between T1 (maximal exertion) and T0 (baseline before exercise) in male ME/CFS patients. * indicates statistically significant changes in cell counts.

T1 vs. T0 in Male ME/CFS Patients		
Cell Type	*p*-Value	Fold Change
CD4+ Naive T cells	0.050 *	−1.372
NK cells	0.019 *	1.529

**Table 3 ijms-24-10255-t003:** Cell type abundance from CIBERSORTx [15] analysis between T2 (4-h after maximal exertion) and T1 (maximal exertion) in healthy controls. * indicates statistically significant changes in cell counts.

T2 vs. T1 in Male ME/CFS Patients		
Cell Type	*p*-Value	Fold Change
CD4+ Naive T cells	0.002 *	1.622
NK cells	0.008 *	−1.461
Dendritic cells	0.023 *	−1.587
Eosinophils	0.002 *	−4.066
**T2 vs. T1 in male HCs**		
CD8+ T cells	0.001 *	12.42
CD4+ Naive T cells	<0.001 *	−2.133
NK cells	0.0003 *	1.817
Dendritic cells	0.002 *	2.171
Eosinophils	<0.001 *	4.063

**Table 4 ijms-24-10255-t004:** Cell type abundance from CIBERSORTx [15] analysis between T2 (4-h after maximal exertion) and T0 (baseline) in male HCs and male ME/CFS patients. * indicates statistically significant changes in cell counts.

T2 vs. T0 in Male ME/CFS Patients		
Cell Type	*p*-Value	Fold Change
CD4+ Memory T cells	0.002 *	1.622
Dendritic cells	0.116	−1.659
Eosinophils	0.050 *	−2.731
**T2 vs. T0 in male HCs**		
CD4+ Naive T cells	0.002 *	1.715
Dendritic cells	<0.001 *	−2.619
Eosinophils	0.002 *	−3.522

## Data Availability

Raw data has been deposited at Gene Expression Omnibus with the accession GSE227375.

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
