# Peer review of "Sex-Dependent Transcriptional Changes in Response to Stress in Patients with Myalgic Encephalomyelitis/Chronic Fatigue Syndrome: A Pilot Project"

_ijms, 2023, doi:10.3390/ijms241210255_

Round 1
Reviewer 1 Report
please see the attachment file

Author Response
Point 1: A first question regards the patient enrolment and statistical stratification, as, while females showed a certain difference in the ShortForm 36-item Survey (SF-36) questionnaire compared to controls, albeit statistical p was not significant, males were reported as having the same "weight". The question is: is it important to be stated for readers?
Response 1: Thank you for your insightful comments on our manuscript. We appreciate your feedback and are glad to have the opportunity to address your concerns. In regards to your concern about male patients with ME/CFS having the same weight, in our study design we matched ME/CFS patients and healthy controls in terms of age and BMI in order to reduce confounding variables and to increase the statistical power of our analysis. Therefore, we believe that it is important to be stated for the readers in order to show that the age and BMI were similar between groups.
Point 2: Table 1: why only males? A sound response is mandatory.
Response 2: We appreciate your comments regarding the data that we did not include in our study. We would like to clarify that the SF-36 data and participant characteristics of female ME/CFS patients and female healthy controls utilized in this study has already been published in the paper titled “Stress-Induced Transcriptomic Changes in females with Myalgic Encephalomyelitis/Chronic Fatigue Syndrome Reveal Disrupted Immune Signatures”. It is reference number 12 in the original manuscript. We intentionally excluded this data that we have already published to avoid redundancy.
In this manuscript, our goal was to extend our analysis to males with ME/CFS and to focus on the sex differences found in transcriptomics of circulating immune cells. However, we acknowledge that our decision not to include this data was not explicitly explained in the manuscript, and we amended the text to address this omission. We included a sentence in section 2.1 in revised manuscript that “these results have been published” with the reference to the aforementioned paper.
We are grateful for your constructive feedback, which will help us to improve the clarity and transparency of our work.
Point 3: Explain how IL-12 is novel in transcriptomic studies and why it was only found in male ME/CFs patients.
Response 3: Thank you for your comments regarding IL-12 as it relates to ME/CFS. We agree that IL-12 has been previously studied in ME/CFS patients at the protein level. After an extensive literature search, we believe that our analysis provides novel insights into the transcriptomics (RNA level) of this pathway that has never before been described. Specifically, our study identifies several genes in the IL-12 pathway that were differentially expressed in ME/CFS male patients as compared to male healthy controls, including genes involved in the production and signaling of IL-12. However, we acknowledge the need to clarify that these findings are novel on the level of transcriptomics (RNA) while also addressing the work done by previous research studies identifying changes in IL-12 at the protein level. In order to rectify this, we added a sentence that “previous research has shown that ME/CFS patients have an increased protein expression value of IL-12p40” and have cited Roerink et al., 2017.
Thank you for your valuable feedback and allowing us to help clarify why these changes in the IL-12 pathway are seen in males only. Numerous studies have reported gender differences in the prevalence, symptoms, and severity of ME/CFS, with many indicating that symptoms are generally worse in females. While the exact role of IL-12 in ME/CFS is not yet known, we have developed two hypotheses to explain why IL-12 pathways expression is increased in males with ME/CFS but not in females with ME/CFS. First, it is possible that the increase in IL-12 seen after exercise-induced stress in males with ME/CFS may be contributing to the disease as male healthy controls did not show these changes and there is ample research showing that increased levels of cytokines contribute to fatigue and pain.
Another hypothesis is that this immune pathway may be contributing to why males with ME/CFS do not experience as severe symptoms as females. Our previous paper, titled “Stress-Induced Transcriptomic Changes in females with Myalgic Encephalomyelitis/Chronic Fatigue Syndrome Reveal Disrupted Immune Signatures”, demonstrated that female healthy controls also showed increased levels of IL-12 after exercise-induced stress as compared to females with ME/CFS. Thus, these changes in gene expression that are similar between males with ME/CFS and healthy female controls after exercise-induced stress may contribute to why males with ME/CFS have fewer symptoms.
In addition, we would like to emphasize that our study is a pilot project. Therefore, we tried to be more conservative in drawing conclusions. Further investigation is needed to validate either of these hypotheses and to elucidate the underlying mechanisms.
Point 4: The authors are invited to discuss the role of NK activity and their number in the research evidence.
Response 4: Thank you for your insightful comments regarding the immunology surrounding NK cells and ME/CFS.
We want to clarify that the increase in NK cells that we showed was derived from RNA-seq data using CIBERSORTx software. CIBERSORTx utilizes an algorithm to infer cell type proportions from gene expression data. We did not assay NK cells using flow cytometry. Therefore, the increases in NK cells that we show are based on the transcriptomic profiles from the PBMCs isolated from the patients. We believe that CIBERSORTx is a valuable tool for identifying this type of data and has been widely used in the field to infer the relative abundance of immune cells in various diseases. While we acknowledge that CIBERSORTx has limitations, such as the assumption of a linear relationship between gene expression and cell proportions, we believe that this data strengthens our discussion on NK cells in ME/CFS patients.
In regards to your thoughtful comments about the role of NK cell activity in our paper, we have added several sentences regarding how our data shows reduced NK lysis and impairments in the NK cell cytotoxic pathways utilizing our gene expression data as well as referencing previous studies that have described impaired NK cell cytotoxicity.
Point 5: Why did the authors not discuss about the role of estrogen receptors in their regulation of IL-12 pathway
Response 5: Thank you for taking the time to provide us with such an insightful comment that will add to the robustness of our discussion involving the role of IL-12 in ME/CFS.
In order to address this, we have added a discussion in section 3.1 describing the role of estrogen in IL-12 production. Further, we have described how estrogen receptor expression may play a critical role in determining IL-12 levels produced in the disease and how this may have implications for understanding the sex differences observed in ME/CFS.
We want to thank you again for your valuable feedback that has served to strengthen our paper.

Reviewer 2 Report
There are several major criticisms of the paper. 1) there is no analysis between T0 and T2. It would seem very important to include this data as it may show the genes most involved in PEM, as PEM may occur up to 48 hours after the exercise challenge. Do we know which cases developed PEM and did any of the gene changes correlate with the severity of that PEM. 2) It is important to show the frequency of the upregulated genes. ME/CFS, according to the currently available data is heterogeneous in etiology and therefore the gene changes identified may only have occurred within a % of the cases. 3) even though the software identified gene pathways having changes examination of the actual genes in the supplementary data shows only part of the pathway is changed. If this is the case it needs to be discussed.
Author Response
Point 1: There is no analysis between T0 and T2. It would seem very important to include this data as it may show the genes most involved in PEM, as PEM may occur up to 48 hours after the exercise challenge.
Response 1: Thank you for this valuable feedback. We have added this analysis to our manuscript in the results (along with figures and tables) and in our discussion. We believe this serves to strengthen our paper and we want to thank you for this helpful input.
Point 2: Do we know which cases developed PEM and did any of the gene changes correlate with the severity of that PEM?
Response 2: Thank you for the opportunity to clarify this point. We did not record whether or not the patients developed PEM. This project was part of a larger study aimed at understanding the biological mechanisms underlying ME/CFS. This study was designed in a way that participants have to pedal to exhaustion. Therefore, they would be sufficiently tired so that the changes in genes and pathways that we see at T2 would be reflective of exercise-induced stress.
In addition our cohort is small and it is a pilot study and we tried to be cautious and conservative in our conclusions. More research is needed with larger cohorts to correlate gene changes with the severity of PEM.
Point 3: It is important to show the frequency of the upregulated genes. ME/CFS, according to the currently available data, is heterogeneous in etiology and therefore the gene changes identified may only have occurred within a % of the cases.
Response 3: Thank you for your valuable feedback regarding the frequency of gene change of our identified differentially expressed genes. We have included a section in each supplementary table that shows the percentage of ME/CFS patients that showed the change in gene expression for each differentially expressed gene.
Point 4: Even though the software identifies gene pathways having changes, examination of the actual genes in the supplementary data shows only part of the pathway is changed. If this is the case it needs to be discussed.
Response 4: Thank you for your insightful comment about the Metascape software. We appreciate the opportunity to provide additional information and clarification how the algorithm works and how it contributes to the biological insights of our study.
In this study, we utilized Metascape because it allowed us to explore and interpret our large scale transcriptomic data using the most updated data sources. We were able to input a list of genes that met the criteria for differential expression in our study into the powerful algorithm. The software then compared them with publicly available datasets to generate insights into the biological processes that are underlying the changes in gene transcripts. Although not every gene that is involved in the pathway may be represented in the gene list that is input into the system, Metascape is able to show which biological functions, pathways and processes are enriched.
We acknowledge that this is a limitation of utilizing the Metascape analysis as any other software for functional analysis, we have made sure to clarify this in our methods section by elaborating that Metascape takes gene lists that are input into it and those are significantly overrepresented in the input gene list are reported to the user as biological pathways that are enriched.

Reviewer 3 Report
The manuscript is devoted to a current topic - studies of the pathophysiology of ME/CFS depending on the gender of the patients. The obtained and analyzed data are original and interesting for specialists in this field and others working in biomedicine. Despite the fact that the material is quite complicated, the manuscript is well planned and written in good language, with explanatory illustrations and tables. However, there are also flaws in the manuscript that need to be addressed.
Comments:
Line 148, Line 324 - Supplementary Table 2 cannot be found because it is not attached;
Line 153, Line 314 - Supplementary Table 3 cannot be found because it is not attached;
Lines 211-212 – “Nodes are colored by ratios between male ME/CFS patients and female ME/CFS patients …” instead of “… male HCs and Female HCs.”;
Line 218 - Supplementary Table 4 is not attached, instead 3 other Tables have been added – Table 4a two with different names “Nanostring validation data for ME/CFS males at T1 v. T0” and “Nanostring validation data for ME/CFS males at T2 v. T1” and Table 4b “Nanostring validation data for male HCs at T2 v. T1”;
Line 235 - Supplementary Table 1 cannot be found because it is not attached;
Line 392 – Figure 4 cannot be found, there are only 3 Figures in the manuscript;
Line 423 – one of the exclusion criterions mentioned is autoimmunity. What did the authors mean by this, since ME/CFS is also considered an autoimmune disease (see references):
Ryabkova VA, Gavrilova NY, Poletaeva AA, Pukhalenko AI, Koshkina IA, Churilov LP, Shoenfeld Y. Autoantibody Correlation Signatures in Fibromyalgia and Myalgic Encephalomyelitis/Chronic Fatigue Syndrome: Association with Symptom Severity. Biomedicines. 2023 Feb; 11(2): 257. Published online 2023 Jan 18. doi: 10.3390/biomedicines11020257
Sotzny F, Blanco J, Capelli E, Castro-Marrero J, Steiner S, Murovska M, Scheibenbogen C; European Network on ME/CFS (EUROMENE). Myalgic Encephalomyelitis/Chronic Fatigue Syndrome - Evidence for an autoimmune disease. Autoimmun Rev. 2018 Jun;17(6):601-609. doi: 10.1016/j.autrev.2018.01.009. Epub 2018 Apr 7.
Author Response
Point 1: Supplementary tables were not able to be viewed and were incorrectly labeled
Response 1: Thank you very much for your thoughtful comments on our manuscript. We appreciate the time and effort that you put into your edits, especially in catching our errors. We have carefully considered your feedback and have incorporated the changes you suggested.
Specifically on our resubmission, we have made sure that the supplementary tables (including supplementary table 1, 2, 3, 4, 5 and 6a, 6b, 6c, 6d and 6e) have been attached and will be available for your viewing.
Point 2: Figure references in paper are incorrect and figure labels need to be changed.
Response 2: We want to apologize for the error in referencing figures. We have corrected this. We have also changed the Figure 3 labels to reflect the changes that you suggested.
Point 3: One of the exclusion criterions mentioned is autoimmunity. What did the authors mean by this, since ME/CFS is also considered an autoimmune disease
Response 3: Thank you for this valuable feedback regarding the exclusion criteria for the study. We would like to clarify that these exclusion criteria were established as part of a larger study aimed at understanding the biological mechanisms of ME/CFS. Although you are correct that autoimmune mechanisms may play a role in the development of the ME/CFS, we were limited by the criteria that was established. We acknowledge that our exclusion criterion may have limited the generalizability but we believe that the benefits of ensuring internal validity of our study outweighed the potential drawbacks of excluding a subset of patients.

Reviewer 4 Report
I would congratulate with authors for the very good paper evaluating differential gene expression by RNA-sequencing (RNA-Seq) in ME/CFS patients and matched healthy controls (HCs) during and after an exercise challenge intended to provoke PEM. Results are extremely interesting, I have only one minor points in order to improve the manuscript. Among exercise challenge authors should also clarify the importance of exercise-induced remodeling as a potential adaptation of cardiac function and structure. In particular, the features of the remodeling may overlap with those of a very early form cardiomyopathies ( DOI: 10.1111/jce.14526 ). Authors should discuss this point for their patient population, more focusing on patient’s characheristic at baseline and should cite suggested reference
Author Response
Point 1: Among exercise challenge authors should also clarify the importance of exercise-induced remodeling as a potential adaptation of cardiac function and structure. In particular, the features of the remodeling may overlap with those of a very early form of cardiomyopathies.
Response 1: Thank you for your valuable feedback on our manuscript. We are truly grateful for your kind words and constructive criticism. We especially appreciate your generosity in sharing your expertise. In light of your comments, we have added a discussion about the importance of correlating transcriptomic results with physiological adaptations during and after exercise in ME/CFS patients. Specifically, the importance of understanding how exercise-induced remodeling on various organs such as the cardiac system can be maladaptive for these patients and overlap with various pathological states. We believe that these clarifications will help improve the overall impact of our manuscript, and we thank you for bringing this important information to our attention.

Round 2
Reviewer 2 Report
This paper is a pilot study and should be more about the mechanisms of the method and the requirements of the final studies if recommended, not a prolonged analysis of the genetic findings on the small sample size. Gene studies are notorious for finding one set of results with a small sample size which are not able to be reproduce with the appropriate sample size. I would liked the discussion to address the issue as well as the adding measures of PEM and recommending an assay at 48 hours after the exercise event.
Author Response
Point 1: This paper is a pilot study and should be more about the mechanisms of the method and the requirements of the final studies if recommended.
Response 1: Thank you for your valuable feedback on our manuscript. We appreciate your comment regarding the need for better implementation of this sex-specific research methods for larger studies. We included a more comprehensive description of potential implications for future research in the discussion section of the manuscript.
We addressed your concern about the small sample size by emphasizing the limitations of the current study and highlighting the need for future research with a larger sample size to validate the findings.
Point 2: I would liked the discussion to address the issue as well as the adding measures of PEM and recommending an assay at 48 hours after the exercise event.
Response 2: We appreciate your input regarding adding measures of PEM and recommending an assay 48 hours after the exercise challenge. Unfortunately, in our study (as in other “omics” studies including exercise) PEM was not assessed after the end of exercise. However, we agree that this is a very valuable measure. We included a recommendation for the need for these measures in future studies utilizing “omics” analysis at various stages of exercise challenge.
Once again, thank you for your valuable feedback, which helped us improve the manuscript and ensure it meets the high standards of the journal.
